# Seroprevalence of Toxoplasmosis among Shelter-Housed Felines in a Philadelphia Suburb

**DOI:** 10.3390/ani12162012

**Published:** 2022-08-09

**Authors:** Danni J. Mitchell, Chelsea L. Reinhard, Stephen D. Cole, Darko Stefanovski, Brittany Watson

**Affiliations:** 1Department of Clinical Sciences and Advanced Medicine, University of Pennsylvania School of Veterinary Medicine, Philadelphia, PA 19104, USA; 2Department of Pathobiology, University of Pennsylvania School of Veterinary Medicine, Philadelphia, PA 19104, USA; 3Department of Clinical Studies—New Bolton Center, University of Pennsylvania School of Veterinary Medicine, Philadelphia, PA 19348, USA

**Keywords:** toxoplasma, shelter-housed cats, seroepidemiologic studies, zoonoses, prevalence

## Abstract

**Simple Summary:**

Cats serve as a host for a parasite called *Toxoplasma gondii*. This parasite can infect other animal species, including humans, and, therefore, the study of toxoplasmosis is relevant to both human and animal health. In this study, we analyzed blood samples from 84 shelter-housed cats to determine if they had been exposed to *T. gondii*. Our results revealed that 28.6% of the cats in our study had been exposed to toxoplasmosis. This study serves as a pilot study for further investigation into the rates of toxoplasmosis infection in shelter-housed felines.

**Abstract:**

Members of the Felidae family are the definitive host of the ubiquitous zoonotic parasite *Toxoplasma gondii*. Few studies have been conducted to determine the epidemiology of *T. gondii* in domestic felines within animal shelter populations. The goal of this study was to assess seroprevalence in a limited-admission shelter in the greater Philadelphia area. Serum samples were collected from cats at a shelter in Media, Pennsylvania during the summer of 2018 to assess the proportion of the population that was IgM or IgG seropositive for antibodies against *T. gondii*, using a commercially available ELISA. Out of the 84 cats that were sampled, 24 cats were seropositive, giving a population prevalence of 28.6%. Nine cats were seropositive for IgM, nine were seropositive for IgG, and six were seropositive for both IgM and IgG. Based on our data, we found that a large percentage of this population was seronegative. Although the sample size in this study was limited and prevented us from obtaining statistically significant results, this research can serve as a pilot study for further investigations into the seroprevalence of toxoplasmosis among shelter-housed felines.

## 1. Introduction

*Toxoplasma gondii* is a parasitic protozoan that is found throughout the world, and it has an ability to infect most avian and mammalian species. As a worldwide zoonotic pathogen that causes toxoplasmosis, environmental contamination of *T. gondii*, therefore, poses a global public health risk [1].

While *T. gondii* infects a variety of species, members of the Felidae family serve as the only definitive hosts. Most felines are infected with toxoplasmosis through the ingestion of infective tissue cysts that reside in prey animals. In immunocompetent adult felines that become infected, clinical signs are typically absent. In animals that become clinically affected, signs, such as lethargy, anorexia, fever, and ocular or neurologic abnormalities, can be seen. Symptomatic infections occur most frequently in congenitally infected kittens [1].

After the ingestion of tissue cysts, *T. gondii* undergoes sexual differentiation in the feline intestinal epithelial cells, which gives rise to the oocyst stage. Infected cats are capable of shedding large numbers of oocysts into the environment for a period of one to two weeks. After shedding, oocysts undergo sporulation (a process that takes 1–5 days) and become infectious [1,2]. This infectious form of the organism survives in the environment for an extended period of time, and, therefore, human and animal exposure is possible through contact with contaminated food, water, or soil [3].

*Toxoplasma gondii* is an important opportunistic infectious pathogen of people. Although toxoplasma infection is typically asymptomatic in immunocompetent adults [4], the infection can be life threatening in organ transplant recipients, AIDS patients, and other immunocompromised patients [5]. In addition, toxoplasmosis during pregnancy can result in infection of the fetus, leading to severe neurologic and ocular disease [6]. *T. gondii* has also been found to impact aquatic ecosystems and was identified as a cause of sea otter mortality [7]. Therefore, an understanding of the exposure rates of *T. gondii* in domestic felines can have a global impact on both human and animal health.

Feline seroprevalence rates range from 16% to 80% in the United States, although studies have varied in the types of populations that were sampled and the type of testing that was performed [8]. Although the prevalence of cats with toxoplasmosis and active oocyst shedding is unknown, data on seroprevalence can provide information about the numbers of felines that have the potential to shed oocysts if they become exposed to *T. gondii*.

Little data exists on the seroprevalence of felines that are entering the sheltering system in the United States. A study that was conducted on stray cats in Pennsylvania in 2009 reported that 41 out of 210 cats were positive for *T. gondii* antibodies. This seroprevalence of 19.5% is lower than most reports from the United States; the authors speculate that the cats in the study were likely unwanted pets and had less exposure to toxoplasmosis [9].

Approximately 1.8 million felines entered the shelters that reported statistics to the Shelter Animals Counts database in 2018 [10]. Seroprevalence information could be helpful in risk factor considerations in shelter cat populations and inform protocol development. Investigating prevalence may aid in understanding support practices that may mitigate the risks of toxoplasmosis infection, such as rodent control. It may also assist in evaluating feline housing (individual vs. group) and other husbandry and sanitation practices that may be linked to toxoplasmosis transmission.

We hypothesized that, based on previous studies in different populations, a substantial population of felines would be seronegative for *T. gondii* exposure upon shelter intake. We hypothesized that animal age, as well as intake status would be associated with seropositivity for *T. gondii*. Several studies have shown that seroprevalence increases with age [11]. Studies have also shown that feral or free roaming cats are at an increased risk of being seropositive [12,13], most likely due to the ingestion of infectious tissue cysts that reside in prey animals [14]. Therefore, cats taken into the shelter system as strays may be at an increased risk for seropositivity.

There were two aims for this study: (1) to determine the proportion of felines in the shelter setting that have been previously exposed to *T. gondii* by antibody detection and (2) to assess whether certain risk factors were associated with a higher rate of *T. gondii* exposure in shelter felines.

## 2. Material and Methods

### 2.1. Serology

Blood samples were collected from domestic felines using a convenience sampling strategy, with samples being collected within 4 months of intake at a shelter in Media, Pennsylvania between June and August 2018. Exclusion criteria included estimated age of less than one year and temperament precluding sample collection. Cats were manually restrained, and a maximum of 3 mL of blood was collected from the medial saphenous vein using a 24 g needle on a 3 mL syringe. The blood was transferred to BD Vacutainer red top tubes and was centrifuged at 1500 rpm for 5 min. The serum was transferred to 1.5 mL Eppendorf tubes and stored at −80 °C until sample analysis. The medical record number of each cat was recorded when the blood sample was collected. Medical records were reviewed by an investigator to record information about animal sex, neuter status, age, state of origin, and intake status. This study and its methods were approved by the University of Pennsylvania Institutional Animal Care and Use Committee (protocol #806518).

Serum samples were analyzed for IgG and IgM antibodies using a reference laboratory-developed enzyme linked immunosorbent assay (ELISA) [15,16]. Sample analysis was performed at Colorado State University Veterinary Diagnostic Laboratories. A titer of 1:64 or higher was considered a positive result.

### 2.2. Population Parameters

The sampled population consisted of 84 felines, with a median age of 3 years (range 1–12 years) (Table 1). Age was not normally distributed. A total of 39% of the population was male, and 71% of the cats had been spayed or neutered prior to shelter intake. Approximately half of the cats (51%) were transferred in from other shelters. The other half of the population was almost evenly distributed between having an intake status of stray (12%), owner surrender (13%), returned adoption (13%), or seized (11%). A total of 89% of the cats were from the Philadelphia tri-state area, and the remaining cats originated from five other states, including Kentucky, Georgia, Florida, Mississippi, and West Virginia.

### 2.3. Statistical Analysis

All analyses were conducted using Stata 16 MP (StataCorp, College Station, TX, USA), with two-sided tests of hypotheses and a *p*-value < 0.05 as the criterion for statistical significance. Tests of normal distribution were performed to determine extent of skewness using the Shapiro–Wilk test. Continuous variables were summarized as medians and ranges. Frequency counts and percentages were used for categorical variables. For the purpose of inference statistics, the χ2 test was used to assess the association between the outcomes of interest (IgG, IgM, and combined IgG and IgM) and categorical independent variables (sex, age, neuter status at the time of intake, intake status, and state of origin).

To estimate the sample size required to detect IgG/IgM based on the origin of the sample, we used our previous results from a study in 84 animals to assume the proportions of animals where IgG/IgM were present or absent based on the origin. Furthermore, we assumed power = 0.8 and alpha = 0.05. Based on the above outlined prior results and assumptions and using a power calculation software PASS 16 (NCSS LLC, Kaysville, Utah), we estimate the total sample to be *n* = 560, where we will have equal number (*n*1 = *n*2 = 280) of origin 0 and 1 samples.

## 3. Results

Analysis of serum samples by ELISA revealed that 24 out of 84 cats were seropositive for IgG, IgM, or both IgG and IgM antibodies against *T. gondii*, giving a seroprevalence of 28.6% (Table 2). Nine cats were seropositive for IgG, nine cats were seropositive for IgM, and six cats were seropositive for both IgG and IgM.

Due to the limited sample size of this study, we were unable to determine if animal sex, age, neuter status at the time of intake, state of origin, or intake status were associated with seropositivity.

## 4. Discussion

The population that was sampled in this study had a combined seroprevalence of 28.6%, which is consistent with the other studies that have been conducted on domestic felines in the United States [8]. Therefore, the population that we sampled did not indicate that shelter cats are at a higher risk of being seropositive compared to other cat populations. Although we were unable to determine if various factors were associated with seropositivity for either IgM or IgG antibodies in shelter felines due to our inability to reach the sample size that was determined by our power calculation, future studies can be performed to investigate possible risk factors with a larger sample size. This study serves as a pilot case study of a single shelter.

One limitation of this study was that sampling did not occur on the day of intake. Due to shelter logistics and convenience sampling, blood samples were drawn within 4 months of the animal’s intake date. It is possible that some of these seropositive felines were exposed to toxoplasmosis after intake. The cats that were enrolled in the study were fed processed diets, which reduced their risk of infection through predation. In future studies, sampling would be performed at both intake and outcome to evaluate for exposure within the shelter setting.

## 5. Conclusions

This study provides novel data on the seroprevalence of toxoplasmosis in shelter-housed felines. In this study population, 28.6% of cats had quantifiable immunity to toxoplasmosis. This study, while limited in sample size, provides a pilot study for further investigations into the seroprevalence rates of toxoplasmosis among shelter-housed felines. This paper provides information on seroprevalence in a population of shelter cats, which has not been explored in previous literature. Tracking population level disease burden and further research looking at factors associated with *T. gondii* transmission could be beneficial in evaluating risks for human and animal health and welfare.

## Figures and Tables

**Table 1 animals-12-02012-t001:** Population parameters for the 84 cats included in the study.

Sex:	
Male	33/84 (39%)
Female	51/84 (61%)
Median estimated age in years (range):	3 (1–12)
Neuter status:	
Males neutered before intake	29/33 (88%)
Females spayed before intake	31/51 (61%)
Males neutered after intake	4/33 (12%)
Females spayed after intake	20/51 (39%)
Intake status:	
Transfer	43/84 (51%)
Stray	10/84 (12%)
Owner surrender	11/84 (13%)
Returned	11/84 (13%)
Seized	9/84 (11%)
State of origin:	
PA/NJ	75/84 (89%)
Other (KY/GA/FL/MS/WV)	9/84 (11%)

**Table 2 animals-12-02012-t002:** Numerical counts of felines within each risk factor group that were seropositive for IgM, IgG, or both IgM and IgG.

	Seronegative	IgM+, IgG−	IgG+, IgM−	IgM+, IgG+
Total number of animals	60	9	9	6
Sex				
Male	28	2	2	1
Female	32	7	7	5
Median estimated age in years	3	4	2	5
Neuter status				
Neutered before intake	26	1	2	0
Spayed before intake	18	5	5	3
Neutered after intake	2	1	0	1
Spayed after intake	14	2	2	2
Intake status				
Transfer	31	4	5	3
Stray	7	0	2	1
Owner surrender	9	1	1	0
Returned	7	2	1	1
Seized	6	2	0	1
State of origin				
PA/NJ	54	8	8	5
Other (KY/GA/FL/MS/WV)	6	1	1	1

## Data Availability

The dataset is available to view only with direct contact to the corresponding authors.

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
