# Peer review of "Seroprevalence of Toxoplasmosis among Shelter-Housed Felines in a Philadelphia Suburb"

_animals, 2022, doi:10.3390/ani12162012_

Round 1
Reviewer 1 Report
A straightforward study that determined that the prevalence rate of T. gondii in cats in a shelter as 28.6% using serology. What is intriguing is that 9 cats were IgM positive and IgG negative suggesting recent infection. It would have been useful if stool samples were examined for T gondii. The exact ELISA test used should be added to the text (it is not sufficient to just indicate that this was done at a reference laboratory). In addition the reference laboratory should be added to the text.
Author Response
A straightforward study that determined that the prevalence rate of T. gondii in cats in a shelter as 28.6% using serology. What is intriguing is that 9 cats were IgM positive and IgG negative suggesting recent infection. It would have been useful if stool samples were examined for T gondii. The exact ELISA test used should be added to the text (it is not sufficient to just indicate that this was done at a reference laboratory). In addition the reference laboratory should be added to the text.
Thank you for the feedback; we appreciate your time in reviewing this study. Additional information was added about the ELISA test that was used and references were included. “Serum samples were analyzed for IgG and IgM antibodies using a reference-laboratory developed enzyme linked immunosorbent assay (ELISA) [15, 16]. Sample analysis was performed at Colorado State University Veterinary Diagnostic Laboratories.”
We agree that information from fecal samples would be useful. We did attempt to collect fecal samples at the time that serum samples were collected. However, due to frequent cage cleaning by shelter staff, we were only able to collect fecal samples for approximately half of the cats that were included in the study. Since detection of oocysts in fecal samples is low (approx 1% of cats in previous studies), we feel that analysis of fecal samples would be appropriate with a larger sample size.
Reviewer 2 Report
Toxoplasmosis is considered an important food-borne parasitic zoonosis which caused by Toxoplasma gondii, and distribute around the world. Cats served as the host for T. gondii which could infect humans as well. In this manuscript, the seroprevalence of T. gondii infection in shelter housed cats in Philadelphia was investigated. The manuscript is well written, while it is better to write materials and methods in sections, as well as discussion. I recommend its acceptance for publication after minor reversion.
1. Keywords: “cat” should be changed to “shelter housed cat”. It is better to remove “parasite”.
2. I suggest that move the sentence “There were two aims for this study: (1) to determine the proportion of felines in the shelter setting that have been previously exposed to T. gondii by antibody detection and (2) to assess whether certain risk factors were associated with a higher rate of T. gondii exposure in shelter felines” to the end of this paragraph.
3. It is suggested that change “Methods” to “Materials and methods”, and move the related information of sample collection from results to this section. The “p” in “p-value” should be italic.
4. Simplify the content of the conclusion and increase the content of the discussion will be better.
5. Reference. Toxoplasma gondii should be italic. The numbers of references are duplicate which should be modified.
Author Response
Toxoplasmosis is considered an important food-borne parasitic zoonosis which caused by Toxoplasma gondii, and distribute around the world. Cats served as the host for T. gondii which could infect humans as well. In this manuscript, the seroprevalence of T. gondii infection in shelter housed cats in Philadelphia was investigated. The manuscript is well written, while it is better to write materials and methods in sections, as well as discussion. I recommend its acceptance for publication after minor reversion.
- Keywords: “cat” should be changed to “shelter housed cat”. It is better to remove “parasite”.
I have updated the key words: “Toxoplasma, shelter housed cats, seroepidemiologic studies, zoonoses, prevalence”.
- I suggest that move the sentence “There were two aims for this study: (1) to determine the proportion of felines in the shelter setting that have been previously exposed to T. gondii by antibody detection and (2) to assess whether certain risk factors were associated with a higher rate of T. gondii exposure in shelter felines” to the end of this paragraph.
The sentence about the study aims has been moved to the end of the introduction as suggested.
- It is suggested that change “Methods” to “Materials and methods”, and move the related information of sample collection from results to this section. The “p” in “p-value” should be italic.
The population parameters have been moved to the “Materials and methods” section, and “p-value” is now formatted correctly.
- Simplify the content of the conclusion and increase the content of the discussion will be better.
The last paragraph was left as the conclusion, with the remaining text being moved into the discussion section.
- Reference. Toxoplasma gondii should be italic. The numbers of references are duplicate which should be modified.
The references have been edited as suggested.
Thank you for your comments; we appreciate your time in reviewing this study.
Reviewer 3 Report
The authors report Toxoplasma antibodies in cats from a humane center in USA. Proper methods were used, and the findings are of interest to scientific community because cats are the only hosts that can disseminate this parasite. The literature review is adequate. However, it will be of interest for authors to discuss the previous survey in 210 shetered cats from Pennsylvania (Dubey et al., J.Parasite 95:578-580, 2009). It is probably worth adding that the cats were most likely pets/indoors and they likely were seropositive before entry to the shelter because in the shelter cats were fed processed food. In reference 8—there is a third edition of the book that enlists all surveys in cats from 2009-2022.
Author Response
The authors report Toxoplasma antibodies in cats from a humane center in USA. Proper methods were used, and the findings are of interest to scientific community because cats are the only hosts that can disseminate this parasite. The literature review is adequate. However, it will be of interest for authors to discuss the previous survey in 210 shetered cats from Pennsylvania (Dubey et al., J.Parasite 95:578-580, 2009). It is probably worth adding that the cats were most likely pets/indoors and they likely were seropositive before entry to the shelter because in the shelter cats were fed processed food. In reference 8—there is a third edition of the book that enlists all surveys in cats from 2009-2022.
Thank you for your comments; we appreciate your time in reviewing this study. We did not come across the article about cats from Pennsylvania in our literature search, so it is a welcome addition to the paper. To the introduction I have added “A study that was conducted on stray cats in Pennsylvania in 2009 reported that 41 out of 210 cats were positive for T. gondii antibodies. Their seroprevalence of 19.5% is lower than most reports from the United States; the authors speculate that the cats in the study were likely unwanted pets, and thus had less exposure to toxoplasmosis.”
We agree that most cats were likely seropositive before coming into the shelter, although we did have some cats that were positive IgM, indicating recent infection. In future studies, it would be ideal to sample cats at intake. This sentence was added to the discussion “The cats that were enrolled in the study were fed processed diets, which reduced their risk of infection through predation”
Thank you for pointing out the new edition- the literature review was written before the new edition was published. This reference was updated.